# Hyperthermia Increases Neurotoxicity Associated with Novel Methcathinones

**DOI:** 10.3390/cells9040965

**Published:** 2020-04-14

**Authors:** Xun Zhou, Jamal Bouitbir, Matthias E. Liechti, Stephan Krähenbühl, Riccardo V. Mancuso

**Affiliations:** 1Division of Clinical Pharmacology & Toxicology, University Hospital Basel, 4031 Basel, Switzerland; xun.zhou@unibas.ch (X.Z.); jamal.bouitbir@unibas.ch (J.B.); matthias.liechti@usb.ch (M.E.L.); riccardo.mancuso@unibas.ch (R.V.M.); 2Department of Biomedicine, University of Basel, 4031 Basel, Switzerland; 3Swiss Centre for Applied Human Toxicology, 4031 Basel, Switzerland

**Keywords:** autophagy, hyperthermia, methcathinone, mitochondria, neurotoxicity

## Abstract

Hyperthermia is one of the severe acute adverse effects that can be caused by the ingestion of recreational drugs, such as methcathinones. The effect of hyperthermia on neurotoxicity is currently not known. The primary aim of our study was therefore to investigate the effects of hyperthermia (40.5 °C) on the neurotoxicity of methcathinone (MC), 4-chloromethcathinone (4-CMC), and 4-methylmethcathinone (4-MMC) in SH-SY5Y cells. We found that 4-CMC and 4-MMC were cytotoxic (decrease in cellular ATP and plasma membrane damage) under both hyper- (40.5 °C) and normothermic conditions (37 °C), whereby cells were more sensitive to the toxicants at 40.5 °C. 4-CMC and 4-MMC impaired the function of the mitochondrial electron transport chain and increased mitochondrial formation of reactive oxygen species (ROS) in SH-SY5Y cells, which were accentuated under hyperthermic conditions. Hyperthermia was associated with a rapid expression of the 70 kilodalton heat shock protein (Hsp70), which partially prevented cell death after 6 h of exposure to the toxicants. After 24 h of exposure, autophagy was stimulated by the toxicants and by hyperthermia but could only partially prevent cell death. In conclusion, hyperthermic conditions increased the neurotoxic properties of methcathinones despite the stimulation of protective mechanisms. These findings may be important for the understanding of the mechanisms and clinical consequences of the neurotoxicity associated with these compounds.

## 1. Introduction

New psychoactive substances (NPSs) are a broad group of drugs of abuse that are not controlled by classic international drug laws [1]. The abuse of NPSs is a major problem worldwide, since NPSs can elicit serious toxic effects on users [2]. In recent years, several synthetic cathinones, designated as “legal highs”, have emerged and their use as recreational drugs has grown rapidly [3]. Structurally, synthetic cathinones are β-keto-amphetamine derivatives, with pharmacological and toxicological properties similar to amphetamines [3]. Synthetic cathinones, such as methcathinone (MC), 4-chloromethcathinone (4-CMC), and 4-methylmethcathinone (4-MMC, mephedrone) (see Appendix A for chemical structures), have recently been recognized by the European Monitoring Centre for Drugs and Drug Addiction (EMCDDA) as emerging NPSs [4,5,6].

Despite clinical studies with and initial use of some synthetic cathinones for the treatment of depression, appetite suppression, or smoking-cessation, none of these compounds have been approved for one of these indications, mainly due to their adverse effect profile [7]. Relevant adverse effects reported for synthetic cathinones include anxiety, paranoia, depression, stroke, seizures, hyperthermia, heart failure, liver failure, and even death [7,8].

Hyperthermia, also reported as “overheating”, is one of the prominent acute severe adverse effects of stimulant drug abuse, and one of the primary causes of death [9,10]. According to clinical case reports, drug-induced hyperthermia can result in many potentially fatal complications, such as hyponatremia, rhabdomyolysis, cerebral edema, disseminated intravascular coagulation, and coma [11]. Drug-induced hyperthermia can be caused by several factors. Most psychostimulant drugs can directly increase metabolic heat production by central and/or peripheral mechanisms as well as decrease heat dissipation [9,10]. Several clinical cases of hyperthermia induced by synthetic cathinones have been reported so far [12] and a large number of animal studies have been performed in mice and rats to investigate the effect of these compounds on the body temperature [13]. Polysubstance abuse may contribute to methcathinone-induced hyperthermia. Additionally, a drug that may accidentally or deliberately be used in combination with cathinones is 3,4-methylenedioxymethamphetamine (MDMA), an amphetamine derivative with well-known effects on thermoregulation [9,14]. In addition, environmental effects that users face in dancing clubs where these drugs are usually consumed may contribute as well as to hyperthermia associated with methcathinones and MDMA [15,16].

While the capacity of cathinones and many other recreational drugs to increase the body and brain temperature is well established, the effects of hyperthermia on neurotoxicity associated with these drugs is currently less well known [13]. Barbosa et al. investigated the effect of ecstasy and ecstasy metabolites on SH-SY5Y cells under normothermic (37 °C) and hyperthermic (40 °C) conditions [17]. They found that the metabolites were more toxic than the parent compound and that the toxicity increased with higher temperature. The aim of the current study was to investigate in vitro the role of hyperthermia on methcathinone-induced neurotoxicity using the well-established SH-SY5Y neuronal cell model [18].

## 2. Materials and Methods

### 2.1. Chemicals and Cell Culture

Amphetamine, 4-fluoroamphetamine (4-FA), methcathinone (MC), 4-fluoromethcathinone (4-FMC), 4-chloromethcathinone (4-CMC), 4-methylmethcathinone (4-MMC), and 3,4-methylenedioxymethamphetamine (MDMA) were purchased from Lipomed (Arlesheim, Switzerland). 4-Chloroamphetamine (PCA) was purchased from Cayman Chemical (Ann Arbor, MI, USA). All drugs were racemic hydrochloride salts with an HPLC purity of >98%. Test drugs were dissolved in dimethyl sulfoxide (DMSO) and stored at −20 °C. The final DMSO concentration during the experiment was 0.1%.

The SH-SY5Y human neuroblastoma cell line was purchased from European Collection of Authenticated Cell Cultures (RRID:CVCL_0019, ECACC) (Sigma-Aldrich, Buchs, Switzerland). SH-SY5Y cells were cultured in a 5% CO_2_ incubator at 37 (normothermic conditions) or 40.5 °C (hyperthermic conditions) in high glucose Dulbecco’s Modified Eagle’s Medium (DMEM) (Thermo Fischer Scientific, Basel, Switzerland) containing 15% heat-inactivated fetal bovine serum (FBS) (Thermo Fischer Scientific, Basel, Switzerland), 2 mM L-glutamine (Thermo Fischer Scientific, Basel, Switzerland), and 1 mM sodium pyruvate (Thermo Fischer Scientific, Basel, Switzerland).

### 2.2. Cell Membrane Toxicity

The release of adenylate kinase (AK) into the cell medium was assessed as a marker to measure plasma membrane integrity. We used the ToxiLight Bioassay Kit (Lonza, Basel, Switzerland) according to the manufacturer’s protocol. In brief, SH-SY5Y cells (50,000 cells per well) were seeded into a 96-well Costar polystyrene plate and left to grow overnight. Afterwards, the medium was removed, and the cells were exposed to different concentrations of PCA (from 100 to 1000 μM), amphetamine, 4-FA, MC, 4-FMC, 4-CMC, and 4-MMC (from 200 to 2000 μM). Triton X-100 (0.1%) was used as a positive control to induce cell lysis. MDMA (500 μM and 1000 μM) was used as a control drug, which is known to induce hyperthermia in vivo [9]. The plate was incubated at 37 and 40.5 °C in 5% CO_2_ and saturated humidity for 6 and 24 h. Then, 20 μL of cell supernatant was transferred into a luminescence-compatible 96-well plate followed by the addition of 50 μL of AK detection reagent. The plate was incubated at room temperature (RT) for 5 min, and the luminescence was measured with a M200 Pro Infinity plate reader (Tecan, Männedorf, Switzerland). All data were normalized to DMSO 0.1%-treated cells (control).

### 2.3. Intracellular ATP Content

Changes in the intracellular ATP content were measured using the CellTiter-Glo^®^ kit from Promega (Dübendorf, Switzerland) according to the manufacturer’s protocol. SH-SY5Y cells were treated as described above. After 6 and 24 h of treatment, 80 μL of assay buffer was added to each well containing SH-SY5Y cells in 80 μL of culture medium. The plate was shaken for 2 min at 350 rpm, followed by 15 min of incubation at RT. The ATP content was determined by luminescence measurement using a M200 Pro Infinity plate reader (Tecan, Männedorf, Switzerland). All data were normalized to DMSO 0.1%-treated cells (control).

### 2.4. Mitochondrial Membrane Potential

The mitochondrial membrane potential (Δψm) was measured using the JC-10 Mitochondrial Membrane Potential Assay Kit (Abcam, Cambridge, UK) according to the manufacturer’s protocol. In healthy cells, JC-10 concentrates in the mitochondrial matrix where it forms red fluorescent aggregates, whereas, in apoptotic and necrotic cells where the Δψm decreases, JC-10 diffuses out of mitochondria, changes to a monomeric form, and stains cells with green fluorescence [19]. SH-SY5Y cells were seeded into black Costar 96-well plates at 50,000 cells per well. Upon incubation with different concentrations of the synthetic methcathinones (200–2000 μM) at 37 or 40.5 °C for 24 h, the supernatant was removed and the cells were rinsed with PBS. Then, 50 μL of JC-10 dye-loading solution was added to each well and the plate was incubated for 15 min at 37 °C and 5% CO_2_ with light protection. Carbonyl cyanide-*p*-trifluoromethoxyphenylhydrazone (FCCP, 100 μM) was used as a positive control. FCCP is an uncoupler of mitochondrial oxidative phosphorylation and therefore decreases Δψm [20]. SH-SY5 cells were exposed to FCCP for 4 h.

The fluorescence was measured using a Tecan M200 Infinite Pro plate reader (Tecan, Männedorf, Switzerland) at 490/525 nm for the aggregates, and at 540/590 nm for monomeric forms. The ratio of the fluorescence intensities between aggregates and monomers was considered as an indicator of Δψm. Data were normalized to control incubations containing DMSO 0.1%.

### 2.5. Mitochondrial Oxygen Consumption

In order to assess the changes in mitochondrial respiration due to hyperthermia in the presence of test drugs, the mitochondrial oxygen consumption rate (OCR) was measured with a Seahorse XF96 analyzer (Seahorse Biosciences, North Billerica, MA, USA). SH-SY5Y cells were seeded at a density of 50,000 cells per well into XF96 Cell Culture Microplates (Seahorse Biosciences, North Billerica, MA, USA) coated with the cell adhesive Corning™ Cell-Tak (22.4 μg/mL) (Corning, New York, USA). SH-SY5Y cells were left to grow overnight and then treated with different concentrations of test drugs (200, 500, and 1000 μM) for 24 h at 37 and 40.5 °C. Before the measurement, SH-SY5Y cells were rinsed twice with unbuffered DMEM medium (4 mM L-glutamate, 1000 μM pyruvate, 1 g/L glucose, and 63.3 mM sodium chloride, pH 7.4) and equilibrated in a CO_2_-free incubator for 30 min. First, basal oxygen consumption was measured, then the leak respiratory rate after automated injection of an ATP synthase inhibitor (oligomycin 1 μM). Maximal OCR was determined by adding FCCP (1 μM). Finally, the non-mitochondrial respiration rate was obtained by the addition of an electron transport chain complex I inhibitor (rotenone 1 μM). OCR was automatically recorded by the Wave software (Seahorse Biosciences, North Billerica, MA, USA), and data were normalized to the protein content determined using the Pierce BCA Protein Assay kit (Thermo Fisher Scientific, Basel, Switzerland). OCR was expressed as pmol O_2_ per minute per mg of protein.

### 2.6. Mitochondrial Superoxide Production

Mitochondrial superoxide production was assessed using the MitoSOX™ Red fluorophore probe (Thermo Fisher Scientific, Basel, Switzerland). SH-SY5Y cells were seeded into a black 96-well plate at a density of 50,000 cells per well, treated with the synthetic methcathinones at different concentrations (from 200 to 2000 μM), and incubated for 24 h at 37 or 40.5 °C. Amiodarone (50 μM) was used as a positive control [21]. MDMA (500 and 1000 μM) was used as a control drug known to induce hyperthermia in vivo [9]. Upon treatment, the medium was removed, and the cells were rinsed with PBS. Next, 100 μL of PBS containing MitoSOX reagent (2.5 μM) was added to each well and the plate was incubated for 10 min at 37 °C with light protection. The fluorescence was measured using a Tecan M200 Infinite Pro plate reader (Tecan, Männedorf, Switzerland) at 510/580 nm. The results were normalized to the protein content quantified by the Pierce BCA Protein Assay kit (Thermo Fisher Scientific, Basel, Switzerland) and to DMSO 0.1%-treated control cells.

### 2.7. Apoptosis

Apoptosis (early and late apoptosis/necrosis) was determined using the Alexa Fluor^®^ 488 annexin V/propidium iodide (PI) staining kit, according to the manufacturer’s protocol (Vybrant TM Apoptosis Assay Kit #2) (Gibco Life Technologies, Paisley, UK) and followed by flow cytometric acquisition using a Cytoflex cytometer (Beckman Coulter, Indianapolis, IN, USA). Briefly, SH-SY5Y cells were seeded into a 48-well plate at a density of 200,000 cells per well and left to grow overnight. The cells were treated with MC, 4-CMC, 4-MMC at 1000 and 2000 μM, and MDMA at 500 and 1000 μM, for 6 and for 24 h at 37 and 40.5 °C. H_2_O_2_ (500 µM) was used as a positive control for apoptosis [22]. The 6-h incubation time was selected to focus on the early apoptosis phase rather than on the necrosis. On the day of measurement, the cells were transferred into a V-well plate, pelleted, and washed twice with PBS. Then, 100 μL of 1× Annexin-Binding Buffer containing 5 μL of Alexa Fluor^®^ 488 annexin V, 1 μL of PI (100 μg/mL), and 0.5 μL of anti-CD29-APC (RRID:AB_314323, clone TS2/16) (BioLegend, San Diego, CA, USA) was added to each well. CD29 is a member of the integrin family expressed on the membrane of SH-SY5Y cells [23]. Thereafter, the plate was incubated at 4 °C for 15 min with light protection. For the flow cytometry gating strategy, singlets were first identified by a forward scatter area (FSC-A) and forward scatter height (FSC-H) gate, and then by an FSC-A and side scatter area (SSC-A) gate. Intact SH-SY5Y cells were distinguished from cell debris by staining with anti-CD29-APC in a FL5-A/SCC-A dot plot. Samples were analyzed using FlowJo software, Tree Star (RRID:SCR_008520) (Ashland, OR, USA).

### 2.8. Western Blotting

SH-SY5Y cells were seeded into 6-well plates and left to grow overnight. For the assessment of Hsp70, caspase 3, LC3 I, and LC3 II, cells were treated with methcathinones (MC, 4-CMC, and 4-MMC) at the concentrations of 1000 and 2000 μM, and with MDMA at 500 and 1000 μM at 37 or 40.5 °C. Concanamycin A (100 nM) and bafilomycin A (100 nM) (Tocris Bioscience, Abingdon, UK) were used as positive controls for the inhibition of autophagy [24]. Hsp70 and caspase 3 were quantified after 6 h of incubation, LC3 I and LC3 II were quantified after 24 h of incubation. Upon treatment, cells were lysed using radioimmunoprecipitation assay (RIPA) buffer with complete protease inhibitor (Roche Diagnostics, Mannheim, Germany) on ice for 15 min and then centrifuged to obtain protein samples. Following the collection of supernatants, BCA Pierce assay was used to quantify the protein concentration of each sample. Afterwards, a total of 18 µg protein per sample was loaded and run onto a 4–12% SDS-PAGE gel and then electroblotted onto a nitrocellulose membrane. The membrane was blocked with 5% non-fat milk in TBST buffer (4 mM Tris base saline containing 0.1% Tween-20, pH 7.5) for 1 h at RT, and then incubated with primary antibody anti-heat shock protein 70 (Hsp70, 1:1000 dilution, ab2787 Abcam, RRID:AB_303300, Cambridge, UK), anti-cleaved caspase 3 (1:500 dilution, ab32042, RRID:AB_725947, Abcam, Cambridge, UK), anti-caspase 3 (1:500 dilution, 8G10, RRID:AB_2069872, Cell Signaling Technology, Danvers, USA), anti-LC3 I/II (1:1000 dilution, 12741s, RRID:AB_2617131, Cell Signaling Technology, Danvers, USA), and anti-glyceraldehyde 3-phosphate dehydrogenase (GAPDH) (1:5000 dilution, sc-365062, RRID:AB_10847862, Santa Cruz Biotechnology, Dallas, USA) overnight at 4 °C. After washing three times with TBST buffer, the membrane was incubated with a secondary antibody (1:2000 dilution, Santa Cruz Biotechnology, USA) for 1 h at RT. Then, signals were developed by the enhanced chemiluminescence (ECL) kit (Bio-Rad Laboratories, Hercules, USA). Protein expression was quantified by the Fusion Pulse TS device (Vilber Lourmat, Oberschwaben, Germany).

### 2.9. Fluorescence Microscopy with Acridine Orange Staining

Acidic vesicular organelles (AVOs) are autolysosomes and autophagosomes and can be measured as markers of the late autophagic process [25,26]. To detect AVOs, cells were stained with acridine orange (AO) according to published protocols [25,26]. AO is a lysosomotropic dye, its fluorescence emission is pH dependent, green at neutral pH and bright yellow to red within acidic organelles [25,26]. SH-SY5Y cells were seeded into ibiTreat µ-Slide (Vitaris, Baar, Switzerland) and exposed to 1000 μM of the synthetic methcathinones for 24 h at 37 °C or 40.5 °C. Then, the cell culture medium was replaced by AO staining solution (0.5 μg/mL acridine orange in culture medium), and the µ-Slide was incubated for 20 min at 37 °C with light protection. The samples were rinsed three times with pre-warmed medium and pictures were acquired by an Olympus IX83 microscope (Olympus, Shinjuku, Japan).

### 2.10. Flow Cytometry with Acridine Orange Staining

To quantify the formation of AVOs, we detected the intensity of AO fluorescence by flow cytometry. AVOs stained with AO emit red fluorescence (FL3-A, 690/50 nm BP); the increase in the intensity of red fluorescence (AO+ events) is proportional to the volume and number of AVOs [25,26]. SH-SY5Y cells were seeded into a 48-well plate at a density of 200,000 cells per well and left to grow overnight. Cells were treated with MC, 4-CMC, and 4-MMC at 1000 and 2000 μM, and MDMA at 500 and 1000 μM, for 24 h, at 37 or 40.5 °C. On the day of measurement, SH-SY5Y cells were transferred and pelleted into a V-well plate, then washed twice with PBS. Afterwards, the cells were incubated with 100 μL of AO staining solution (0.5 μg/mL in PBS) for 20 min at 37 °C with light protection. SH-SY5Y cells were analyzed with a CytoFLEX flow cytometer (Beckman Coulter, IN, USA). For the flow cytometry gating strategy, singlets were first identified by an FSC-A/FSC-H gate, and then by an FSC-A/SSC-A gate. AO+ cells were identified in an SSC-A/FL3-A dot plot, in comparison to DMSO 0.1%-treated cells (control). Results were analyzed using the FlowJo software (Tree Star, Ashland, OR, USA).

### 2.11. Statistics

Experimental data are presented as the mean ± SEM of at least five independent experiments. Statistical comparisons were performed with one-way ANOVA followed by *t*-tests. Statistical significance was considered at *P* ≤ 0.05. GraphPad Prism 8.3.0 (RRID:SCR_002798) (GraphPad Software, La Jolla, CA, USA) was used for all statistical analyses.

## 3. Results

### 3.1. Cell Membrane Integrity and ATP Content

In order to obtain an overview of the effect of hyperthermia on amphetamine- and methcathinone-induced neurotoxicity, we first determined the release of AK and the intracellular ATP content in SH-SY5Y after 24 h of drug exposure under normothermic (37 °C) and hyperthermic conditions (40.5 °C). AK release is commonly used as a marker of cell membrane integrity, whereas the intracellular ATP content represents a marker of energy metabolism. SH-SY5Y cells were exposed to increasing concentrations of amphetamine, 4-fluoroamphetamine (4-FA), 4-chloroamphetamine (PCA), methcathinone (MC), 4-fluoromethcathinone (4-FMC), 4-chloromethcathinone (4-CMC), and 4-methylmethcathinone (4-MMC) (see Appendix A for chemical structures). MDMA was also included due its widespread use and its known effects on body temperature.

As shown in Figure 1 for methcathinones and MDMA and in Appendix A for the amphetamines, all of these compounds were membrane toxic and decreased the intracellular ATP content in a concentration-dependent manner. Exceptions were MC and 4-FMC, which did not show any significant toxicity up to 2000 μM (Figure 1). 4-FA and PCA were membrane toxic starting at 1000 and 500 μM, respectively, at both temperatures investigated (Appendix A), whereas 4-CMC, 4-MMC, and MDMA were significantly more toxic at 40.5 °C, with membrane toxicity starting at 1000 μM at this temperature (Figure 1A).

The intracellular ATP content in SH-SY5Y cells started to decrease at 2000 μM for 4-FA, 4-CMC, and 4-MMC, and at 1000 μM for MDMA at normothermic conditions, whereas at 40.5 °C, it started to decrease at 2000 μM for amphetamine; 1000 μM for 4-FA, MDMA, and 4-MMC; at 200 μM for PCA; and at 500 μM for 4-CMC (Figure 1B and Appendix A). 4-FA, 4-CMC, 4-MMC, and MDMA were significantly more toxic under hyperthermic conditions (Figure 1B and Appendix A), in line with the findings of the AK assessment experiments. Moreover, the drugs investigated showed a more pronounced toxicity regarding the decrease in the intracellular ATP content when compared to membrane toxicity, a pattern suggesting mitochondrial toxicity (Appendix A).

Based on these first screenings, we decided to investigate the effect of hyperthermia on the neurotoxicity associated with the synthetic methcathinones MC, 4-CMC, and 4-MMC in more detail.

### 3.2. Mitochondrial Membrane Potential

In order to understand the mechanism of temperature-increased mitochondrial toxicity, we determined the Δψm by staining SH-SY55 cells with the JC-10 dye [21]. Our data indicated that MC did not change the Δψm significantly up to 2000 µM (Appendix A). Similarly, MDMA was associated with a numeric drop in the Δψm but without reaching statistical significance (Appendix A). In contrast, 4-CMC and 4-MMC decreased the Δψm in a concentration-dependent manner at both temperature conditions (Appendix A), reaching statistical significance at 2000 and 1000 µM, respectively. In contrast to AK release and ATP depletion, the Δψm did not show a more accentuated decrease at 40.5 °C compared to 37 °C.

### 3.3. Mitochondrial Oxygen Consumption

The observed decrease in intracellular ATP and Δψm could be caused by impaired mitochondrial function. We therefore assessed the oxygen consumption rate (OCR) by exposing SH-SY5Y cells for 24 h at 37 and 40.5 °C to the test compounds using a Seahorse XF96 analyzer. While MC was not toxic, 4-CMC (1000 μM), 4-MMC (1000 μM), and MDMA (500 µM) significantly decreased mitochondrial basal and FCCP-stimulated respiration at 37 °C (Figure 2). In comparison, at 40.5 °C, all compounds investigated started to be toxic already at 200 μM for both basal and FCCP-stimulated respiration (Figure 2). We also recorded the leak respiration, which is the cellular uptake of oxygen in the presence of the F_1_F_0_-ATP synthase inhibitor oligomycin. An increase in the leak respiration would indicate uncoupling of oxidative phosphorylation. Since none of the compounds investigated stimulated leak respiration, we can exclude uncoupling as a reason for the observed decrease in Δψm.

In contrast to the effect on the mitochondrial membrane potential, these data showed that hyperthermic conditions clearly increased the mitochondrial toxicity of MC, 4-CMC, 4-MMC, and MDMA. Furthermore, halogenation and methylation in the *p*-position increased the toxicity of the methcathinones in comparison to hydrogen in this position.

### 3.4. Mitochondrial Superoxide Production

To further investigate the potential involvement of mitochondria in the observed toxicity, we determined the mitochondrial ROS production. The inhibition of complex I and III of the electron transport chain can increase the production of mitochondrial superoxide [27,28]. Mitochondrial ROS were measured in SH-SY5Y cells exposed to methcathinones for 24 h at 37 and at 40.5 °C. Concerning normothermic conditions, only 4-CMC showed a significant increase of the mitochondrial ROS content, which started at 2000 μM. At 40.5 °C, the superoxide anion content increased significantly starting at 2000 μM for MC, and at 1000 μM for 4-CMC, 4-MMC, and MDMA (Figure 3).

### 3.5. Cell Death Mechanisms

Inhibition of oxidative phosphorylation with a decrease in the cellular ATP content and intracellular accumulation of ROS could initiate cell death mechanisms, such as apoptosis and/or necrosis [29]. To clarify whether hyperthermia can affect such events in drug-treated SH-SY5Y cells, we analyzed the proportion of apoptotic and necrotic cells after treatment with test compounds at 37 and 40.5 °C for 6 and 24 h.

As expected, H_2_O_2_ at 0.5 μM increased the number of apoptotic and necrotic cells at both temperatures (Figure 4A). At 37 °C, after 6 h of incubation, 4-CMC increased the percentage of apoptotic cells significantly, starting at 1000 μM (Figure 4A). In comparison, the other compounds tested increased the percentage of apoptotic cells only numerically, without reaching statistical significance. 4-CMC and 4-MMC increased the percentage of necrotic cells starting at 1000 and 2000 μM, respectively, and MDMA starting at 1000 µM. Similar results were obtained for caspase 3 activation at 37 °C (Figure 4B). 4-CMC and 4-MMC increased caspase 3 cleavage starting at 1000 µM, whereas the other compounds tested increased caspase 3 cleavage only numerically. In comparison, at 40.5 °C, the effect of the compounds investigated on the percentage of apoptotic cells (Figure 4A) and caspase 3 cleavage (Figure 4B) was less accentuated than at 37 °C. 4-CMC induced apoptosis, necrosis, and caspase 3 cleavage at 2000 µM, whereas the other compounds were not associated with significant increases in the induction of apoptosis, necrosis, or caspase 3 cleavage up to 2000 μM. After 24 h of incubation at 37 °C, none of the substances tested induced apoptosis, but 4-CMC and 4-MMC induced necrosis starting at 2000 µM (Appendix A). At 40.5 °C, the picture was not different from 37 °C, but the extent of necrosis was numerically more accentuated.

The data suggested a shift from apoptosis to necrosis at the higher temperature and with longer incubation. In order to obtain a better understanding of this shift, we also determined the AK release and ATP content in the presence of the methcathinone derivatives after 6 h of incubation (Appendix A). The ATP content dropped, and the AK release increased only at the highest 4-CMC concentrations at both temperatures investigated, which is compatible with the increase in cell necrosis observed with this compound.

To better understand the effects of hyperthermia on the cell death pathways, we measured the expression of Hsp70 protein in SH-SY5Y cells after 6 h of exposure to the synthetic methcathinones. Hsp70 are a family of proteins, which protect the cells from stress by helping cellular proteins to retain their native conformation or to regain their function after misfolding due to an increased temperature. Hsp70 proteins are usually upregulated by heat stress and toxic chemicals [30]. As expected, Hsp70 expression was significantly increased at 40.5 °C compared to 37 °C for all compounds tested as well as for DMSO-treated control cells (Figure 5). In comparison to control incubations, the compounds investigated did not influence Hsp70 expression at 37 or 40.5 °C.

### 3.6. Detection of Autophagy

Autophagy is a physiological catabolic process used by most cells to remove misfolded proteins or entire organelles in order to maintain proteo- and homeostasis. It is triggered mainly by starvation, cell insults, and DNA damage [31]. Hyperthermic conditions, exposure to mitochondrial toxicants, and cellular ROS accumulation can generate misfolded proteins, which can affect the cellular metabolism [32]. In order to explore whether hyperthermia and/or methcathinones can promote autophagy in SH-SY5Y cells, we analyzed the expression of autophagy-related protein microtubule-associated protein 1A/1B-light chain 3 (LC3) by Western blots [25]. LC3 is a component of the autophagosome membranes in mammalian cells, and it is a validated marker for the assessment of early autophagy [33]. During the first phase of autophagy, the cytosolic form of LC3 (LC3 I) is cleaved and conjugated with phosphatidylethanolamine to form LC3 II, which is then translocated into autophagosome membranes. An increase in the LC3 II/LC3 I ratio therefore reflects autophagosome formation [34]. SH-SY5Y cells were exposed at 37 and 40.5 °C to MC, 4-CMC, 4-MMC (1000 and 2000 µM), and MDMA (500 and 1000 µM). After 24 h of incubation at 37 °C, all test compounds showed a concentration-dependent increase in the LC3 II/LC3 I ratio compared to control conditions, which reached statistical significance for 2000 µM 4-CMC and for 2000 µM 4-MMC (Figure 6). Under hyperthermic conditions, the effect of the test compounds on the LC3 II/LC3 I ratio was similar but more accentuated for 4-CMC and 4-MMC as compared to 37 °C.

In order to investigate whether the observed increase in the LC3 II/LC3 I ratio in the presence of 4-CMC and 4-MMC was due to increased phagosome formation or decreased degradation, we investigated the effect of the two autophagy inhibitors concanamycin A and bafilomycin A on the LC3 II/LC3 I ratio. Concanamycin A and bafilomycin A impair the fusion of autophagosomes and lysosomes by inhibiting the vacuolar H^+^ ATPase and therefore the acidification of organelles, which increases LC3 II expression [24]. As shown in Figure 6A, in the absence of concanamycin A and bafilomycin A, treatment with 2000 µM 4-CMC or 4-MMC significantly increased the LC3 II/LC3 I ratio both at 37 and 40.5 °C compared to control incubations, with 40.5 °C being more effective than 37 °C. As shown in Figure 6B, as expected, concanamycin A and bafilomycin A increased the LC3 II/LC3 I ratio at both temperatures compared to the DMSO control. At 37 °C, concanamycin A and bafilomycin A increased the LC3 II/LC3 I ratio in the presence of 4-CMC or 4-MMC compared to the respective control incubations (2000 μM 4-CMC or 4-MMC). At 40.5 °C, this effect was less prominent and did not reach statistical significance. Importantly, the LC3 II/LC3 I ratios of incubations containing only concanamycin A or bafilomycin A and of incubations containing concanamycin A or bafilomycin A in combination with 4-CMC or 4-MMC were not different at both temperatures. Since concanamycin A and bafilomycin A block autosome degradation, the LC3 II/LC3 I ratio in the presence of these inhibitors reflects autophagosome formation, which was obviously not different between incubations containing only concanamycin A or bafilomycin A and incubations containing concanamycin A or bafilomycin A in combination with 4-CMC or 4-MMC. The results therefore suggest that the observed increase in the LC3 II/LC3 I ratio in the presence of 4-CMC and 4-MMC reflects impaired autophagosome degradation and not formation.

To further study the autophagy, we assessed the formation of acidic vesicular organelles (AVOs) [25], which is a hallmark of late autophagy [26]. Microscopic analysis of SH-SY5Y cells exposed for 24 h at 37 and 40.5 °C to MC, 4-CMC, and 4-MMC (1000 μM) showed an accumulation of AO dye within acidic compartments (Figure 7A). All the tested drugs increased the formation and size of the AVOs at both temperature conditions. Moreover, 4-CMC and 4-MMC at hyperthermic conditions affected the morphology of the cells, with a reduction of neuritic processes (Figure 7A). To confirm the effect of the investigated compounds on AVO formation, SH-SY5Y cells were exposed for 24 h at 37 and 40.5 °C to MC and 4-MMC (1000 and 2000 μM) and to 4-CMC (500 and 1000 μM) and AVOs were quantified by flow cytometry (Figure 7B). At 37 °C, 4-CMC and 4-MMC showed a significantly increased percentage of AO+ cells at 1000 μM compared to control incubations. Compared to 37 °C, exposure to 4-CMC and 4-MMC at 40.5 °C was associated with more AVO formation (Figure 7B), and 4-CMC significantly increased AVO abundance at 40.5 °C already at 500 μM.

## 4. Discussion

Barbosa et al. [17] have shown in a previous study that the toxicity of ecstasy and ecstasy metabolites on differentiated SH-SY5Y cells is increased at hyperthermic (40 °C) compared to normothermic (37 °C) conditions. Similarly, in the present study, we demonstrated that hyperthermic conditions increase the cytotoxicity of methcathinones by inducing apoptotic and necrotic cell death. The loss in cell membrane integrity was preceded by a decrease in cellular ATP content, a typical feature of mitochondrial toxicants [21,35]. We could confirm our previous investigations on the hepatocellular and muscle toxicity observed with these drugs [27,36].

The choice of the higher temperature was based on the study of Barbosa et al. [17] and on clinical reports suggesting that body temperatures ≥40 °C are associated with severe complications, such as intravascular coagulation, rhabdomyolysis, renal and multiorgan failure, and even death, in patients having ingested ecstasy or other psychostimulants [9,10].

The mitochondrial toxicity of the methcathinones studied was confirmed by the investigation of their effect on mitochondrial respiration. For all compounds studied, a prominent reduction of basal and FCCP-induced respiration was observed at 40.5 °C compared to 37 °C. Mitochondrial toxicants that reduce the OCR, which indicates an impairment of the electron transport chain, can result in a decrease in Δψm and an increase in mitochondrial generation of ROS [37], which we observed in the current study. In agreement with the results of the current study, previous in vitro investigations have shown that synthetic methcathinones increase the production of ROS and nitrogen reactive species in human dopaminergic SH-SY5Y cells [25]. Moreover, animal studies revealed that methcathinones increase the expression of the antioxidant enzymes superoxide dismutase, catalase, and glutathione peroxidase in response to ROS accumulation [38].

Mitochondrial impairment and ROS accumulation can lead to the opening of the mitochondrial permeability transition pore (mPTP), and subsequently to the release of cytochrome *c* into the cytoplasm, which is followed by caspase activation and apoptosis [39,40]. In our study, exposure to 4-CMC and 4-MMC already induced apoptosis and/or necrosis at both temperature conditions after 6 h of incubation.

Surprisingly, after 6 h of incubation, the compounds investigated were more toxic at normothermic conditions than under hyperthermic conditions, whereas, after 24 h, the temperature shift from 37 to 40.5 °C clearly increased the toxicity of the methcathinones studied. These findings suggested the induction of defensive mechanisms that could protect the cells during the first hours of the caloric insult. We therefore determined the expression of Hsp70 proteins, which are molecular chaperones synthesized in response to a heat shock [41]. As shown in Figure 5, hyperthermic conditions increased the expression of Hsp70 proteins, independently of the exposure to the compounds tested. Already after 6 h at 40.5 °C, the expression of Hsp70 was significantly higher than under normothermic conditions. These proteins, which assist in the folding and assembly of newly synthesized proteins as well as refolding of misfolded and aggregated proteins, act in an ATP-controlled fashion [42]. Importantly, Hsp70 proteins can inhibit mitochondrial cytochrome *c* release, apoptosome formation, and caspase activation [43,44]. The results of the current study suggest that at the 6-h time point, the induction of HSP70 partially prevented not only the cellular damage induced by heat but also the toxic insult by the methcathinones investigated.

However, because of the ongoing ROS-induced protein and organelle damage and the associated decrease in the cellular ATP content, this defensive system may be not able to counteract the cellular insults. Autophagy is an additional defensive mechanism, which may be stimulated under these conditions. In support of this assumption, it has been shown that 3-fluoromethcathinone (3-FMC), another synthetic methcathinone, activates autophagy in the neuronal cell line HT22 [45]. Autophagy is an evolutionarily conserved process, which plays a pivotal role in maintaining the viability of eukaryotic organisms by regulating the intracellular balance between anabolism and catabolism [46]. Autophagy is activated as a protective mechanism when cells are not able to respond to nutrient stress, the accumulation of misfolded proteins, and/or organelle damage [47]. In our investigations, LC3 II, an early marker of autophagy [48], was overexpressed in SH-SY5Y cells after exposure to 4-CMC under both temperature conditions. Furthermore, we demonstrated the stimulation of autophagy directly by the visualization of acidic vesicular organelles (AVOs) in SH-SY5Y cells exposed to 4-CMC and 4-MMC, which is a marker of late autophagy. Similar to the induction of HSP70, hyperthermic conditions also increased the formation of AVOs, suggesting that the activation of the cellular defense systems is tightly coordinated.

In comparison to their pharmacological activity, which is observed in the high nanomolar to micromolar range depending on the compound and the pharmacological effect considered [49], cytotoxicity was detected at higher concentrations in the current study. For 4-CMC, blood concentrations reached approximately 1 micromolar with non-toxic pharmacological doses and up to 10 micromolar in patients with intoxications [50]. In the current study, we started to see an impairment of cellular oxygen uptake at 200 micromolar for MC, 4-CMC, and 4-MMC. A possible explanation for this discrepancy between pharmacological activity and in vitro toxicity may be that the cell lines used in the current study may be less sensitive to toxicants than primary cells. This has, for instance, been shown for hepatotoxicants, which are typically less toxic for human hepatocyte cell lines than for primary human hepatocytes [51,52]. Furthermore, patients presenting with neurotoxicity have usually ingested doses higher than the pharmacological doses of these compounds and may have ingested additional toxic drugs and/or alcohol. Finally, the brain/plasma concentration ratio for 4-MMC is >1, indicating that the drug penetrates the blood–brain barrier easily and that the concentration reached in the brain is higher than in plasma [53].

Interestingly, many studies have shown a link between autophagy and neurodegenerative diseases [31,54,55]. A prevalent pathological feature of many neurodegenerative diseases, such as Alzheimer’s disease (AD), Parkinson’s disease (PD), and Huntington disease (HD), is in fact the aggregation of misfolded proteins, which may result from the production of defective proteins and/or impaired function of the protein quality control systems. Our studies suggest that repetitive ingestion of neurotoxic drugs, such as methcathinones, and other neurotoxic neurostimulants may aggravate or even provoke such conditions.

## 5. Conclusions

In conclusion, 4-CMC and 4-MMC are mitochondrial toxicants whose toxicity is increased by shifting the temperature from 37 to 40.5 °C. SH-SY5Y cells exposed to 40.5 °C activate cellular defense mechanisms, such as the expression of Hsp70 proteins, which can partially prevent early apoptosis and necrosis. With time, the activation of additional defense mechanisms, such as autophagy, is necessary to prevent cell dysfunction and cell death. Mitochondrial toxicity, which is accentuated by hyperthermia, represents an important mechanism of the neural toxicity of these compounds.

## Figures and Tables

**Figure 1 cells-09-00965-f001:**
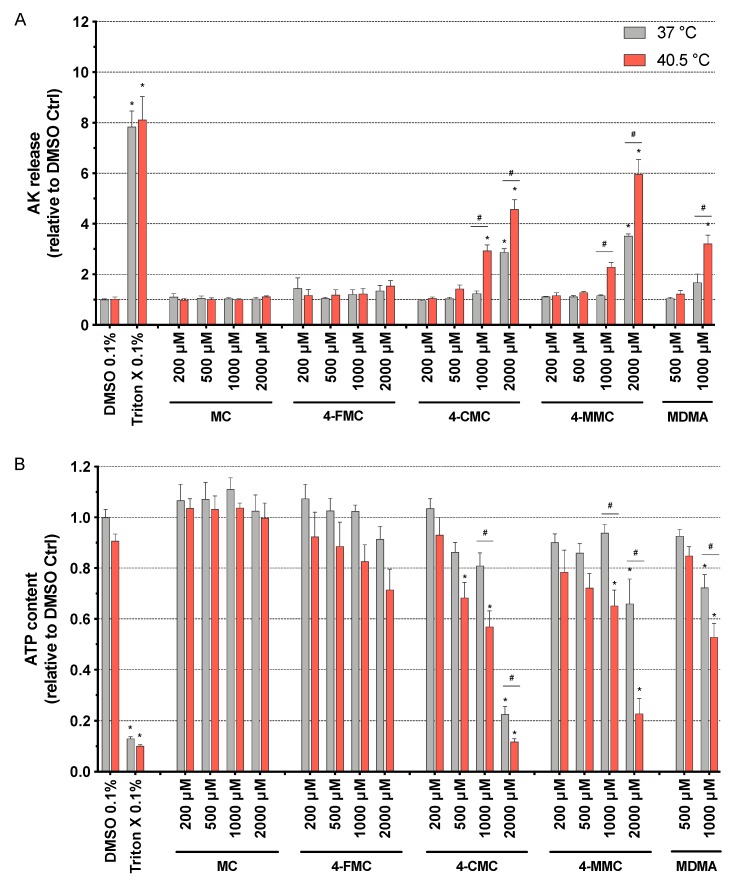
(**A**) Plasma membrane integrity and (**B**) intracellular ATP content assessed in SH-SY5Y cells after 24 h of exposure at 37 and 40.5 °C to methcathinone (MC), 4-fluoromethcathinone (4-FMC), 4-chloromethcathinone (4-CMC), 4-methylmethcathinone (4-MMC) (200-2000 μM), and 3,4-methylenedioxymethamphetamine (MDMA) (500 and 1000 μM). Dimethyl sulfoxide (DMSO) and Triton X were used as negative and positive controls, respectively. Data are expressed relative to the DMSO control as the mean ± SEM of eight independent experiments. Statistical comparisons were performed with one-way ANOVA followed by *t*-tests (**P* ≤ 0.05 versus control at the same temperature; ^#^*P* ≤ 0.05 versus the same concentration at a different temperature).

**Figure 2 cells-09-00965-f002:**
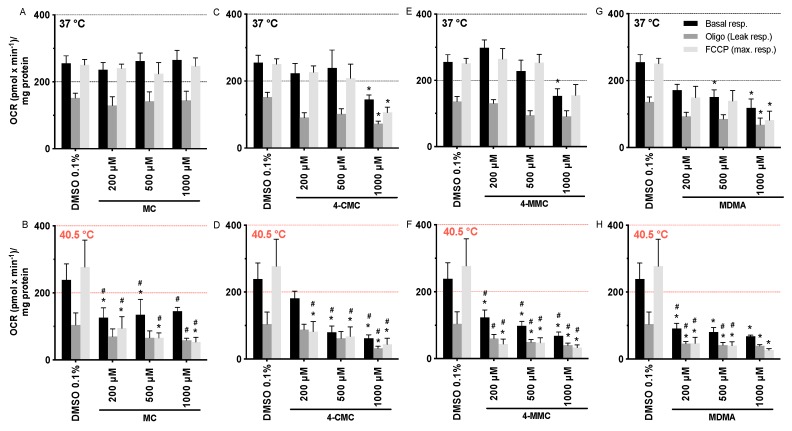
Oxygen consumption rate of SH-SY5Y cells at 37 and 40.5 °C expressed as basal, leak, and maximal respiration. SH-SY5Y cells were exposed for 24 h to (**A,B**) MC, (**C,D**) 4-CMC (**E,F**) 4-MMC, and (**G,H**) MDMA. Data are expressed as mean ± SEM of eight independent experiments. Statistical comparisons were performed with one-way ANOVA followed by *t*-tests (**P* ≤ 0.05 versus control at the same temperature; ^#^*P* ≤ 0.05 versus the same concentration at a different temperature).

**Figure 3 cells-09-00965-f003:**
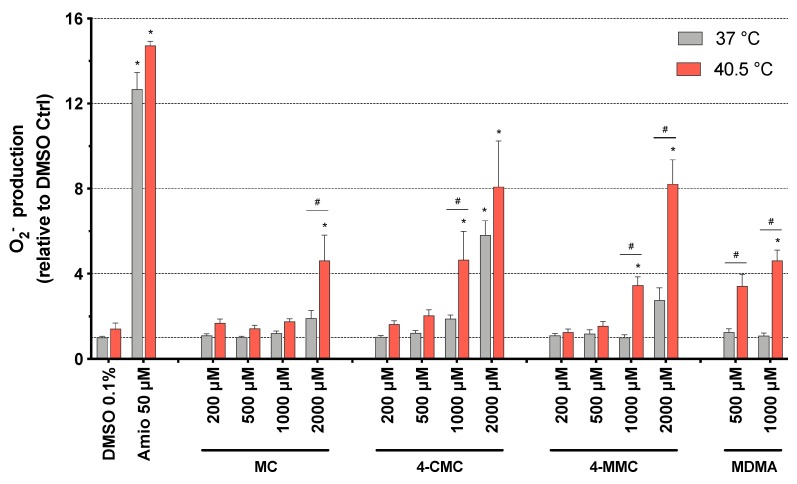
Mitochondrial superoxide production in SH-SY5Y cells at 37 and 40.5 °C after 24 h of exposure to MC, 4-CMC, 4-MMC (200–2000 μM), and MDMA (500 and 1000 μM). DMSO and amiodarone were used as negative and positive controls, respectively. Data are expressed relative to DMSO control as mean ± SEM of six independent experiments run in quadruplicate. Statistical comparisons were performed with one-way ANOVA followed by *t*-tests (**P* ≤ 0.05 versus control at the same temperature; ^#^*P* ≤ 0.05 versus the same concentration at a different temperature).

**Figure 4 cells-09-00965-f004:**
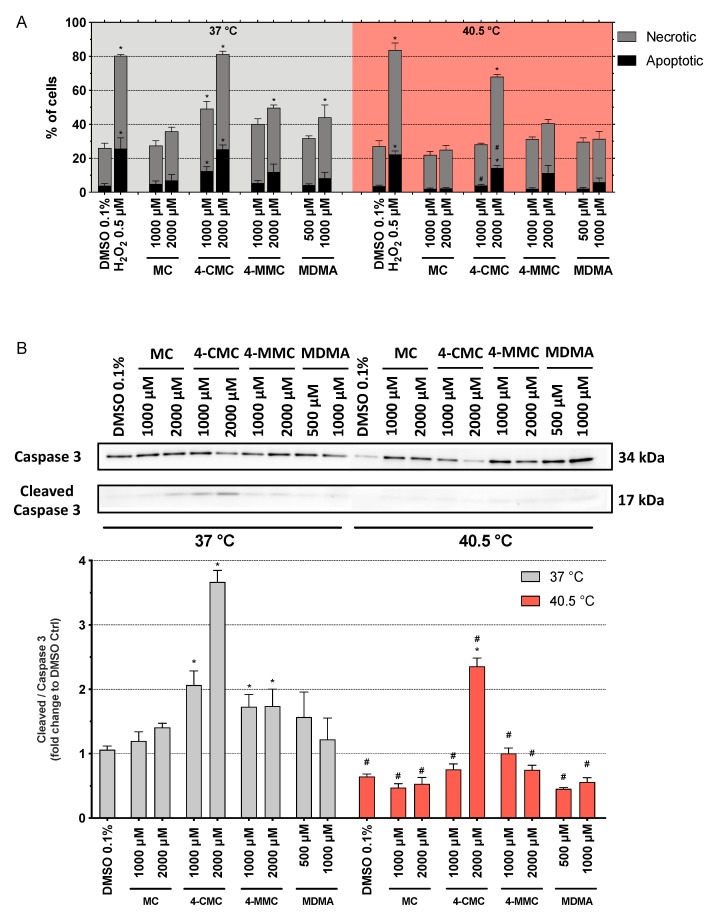
Mechanisms of cell death after 6 h of exposure at 37 and 40.5 °C to MC, 4-CMC, 4-MMC (1000 and 2000 μM), and MDMA (500 and 1000 μM). DMSO and H_2_O_2_ were used as negative and positive controls, respectively. (**A**) Percentage of necrotic and apoptotic cells. (**B**) Activation of caspase 3. Data are expressed as mean ± SEM of six independent experiments. Statistical comparisons were performed with one-way ANOVA followed by *t*-tests (**P* ≤ 0.05 versus control at the same temperature; ^#^*P* ≤ 0.05 versus the same concentration at a different temperature).

**Figure 5 cells-09-00965-f005:**
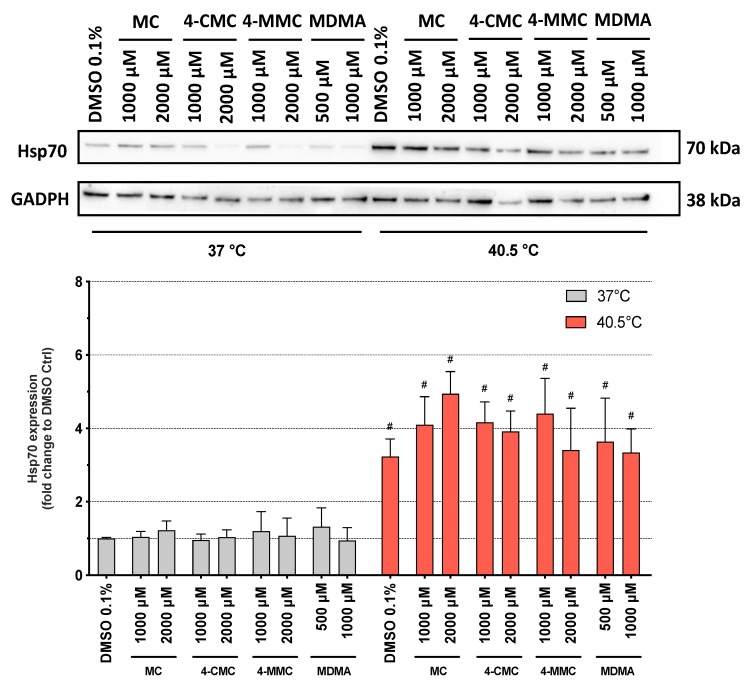
Expression of 70 kilodalton heat shock protein (Hsp70) in SH-SY5Y cells after 6 h of exposure at 37 and 40.5 °C to MC, 4-CMC, 4-MMC (200–2000 μM), and MDMA (500 and 1000 μM). Data are expressed relative to the DMSO control as mean ± SEM of six independent experiments. Statistical comparisons were performed with one-way ANOVA followed by *t*-tests (^#^*P* ≤ 0.05 versus the same concentration at a different temperature).

**Figure 6 cells-09-00965-f006:**
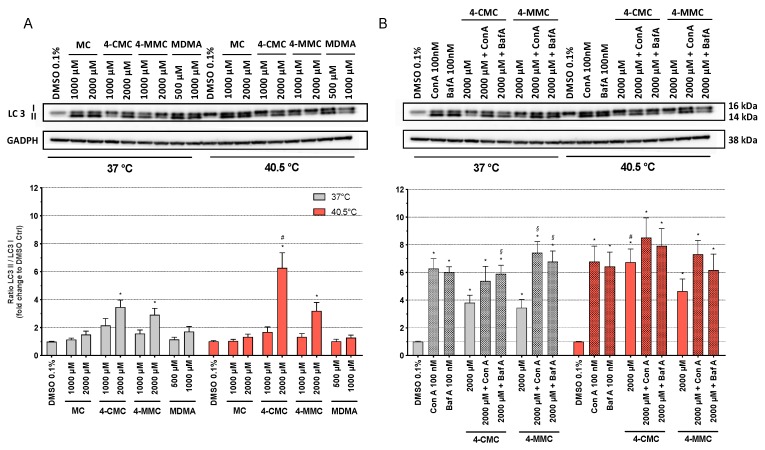
Protein expression of microtubule-associated protein 1A/1B-light chain 3 (LC3) I and LC3 II in SH-SY5Y cells after 24 h of exposure at 37 and 40.5 °C to (**A**) MC, 4-CMC, 4-MMC (200–2000 μM), and MDMA (500 and 1000 μM) and (**B**) in the presence of the autophagy inhibitors concanamycin A (100 nM) and bafilomycin A (100 nM). Data are expressed as the ratio of LC3 II/LC3 I relative to the DMSO control of six independent experiments. Statistical comparisons were performed with one-way ANOVA followed by paired *t*-test (**P* ≤ 0.05 versus DMSO control at the same temperature; ^#^*P* ≤ 0.05 versus the same concentration at a different temperature; *^§^P* ≤ 0.05 versus incubations containing 4-CMC or 4-MMC without concanamycin A or bafilomycin A).

**Figure 7 cells-09-00965-f007:**
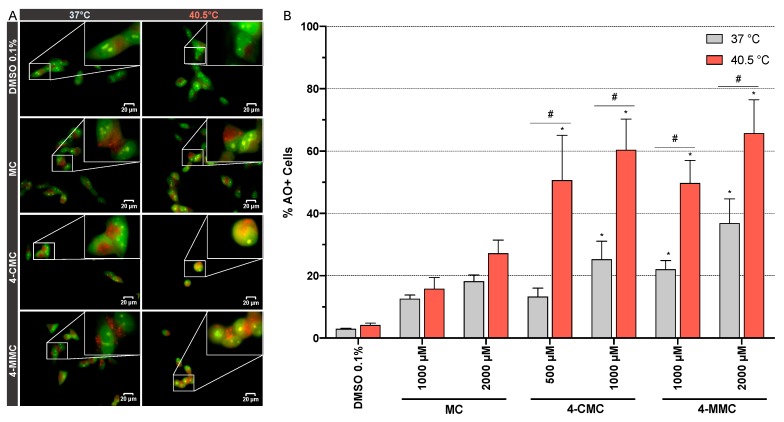
(**A**) Fluorescence microscopy visualization of acidic vesicular organelles (AVOs) stained with acridine orange (AO). SH-SY5Y cells were exposed to MC, 4-CMC, and 4-MMC (1000 μM) for 24 h at 37 and 40.5 °C. Scale bar 20 μm, magnification 60×. (**B**) Flow cytometric quantification of AVOs. SH-SY5Y cells were exposed to MC (1000 and 2000 μM), 4-CMC (500 and 1000 μM), and 4-MMC (1000 and 2000 μM) for 24 h at 37 and 40.5 °C. Data are expressed as mean ± SEM of five independent experiments. Statistical comparisons were performed with one-way ANOVA followed by *t*-tests (**P* ≤ 0.05 versus control at the same temperature; ^#^*P* ≤ 0.05 versus the same concentration at a different temperature).

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
