# Peer review of "Hyperthermia Increases Neurotoxicity Associated with Novel Methcathinones"

_cells, 2020, doi:10.3390/cells9040965_

Round 1
Reviewer 1 Report
The primary of this study was to investigate the effects hyperthermia (40.5 °C) on the neurotoxicity of methcathinone (MC), 4-chloromethcathinon, (4-CMC), and 4-methylmethcathinone (4-MMC) in SH-SY5Y cells.
- This study is well planned however, authors restricted their study to two different temperatures. 37 and 40.5. I would recommend to use additional temp measurements of 38 and 39 as the amount of temperature increase is dose dependent.
- The amount of dose used for in vitro study was very high. Does the author have a reasonable explanation for using such as high dose.
- This article is well written with few edits in grammar and sentence phrasing.
- Line 248 "In contrast to AK release and ATP depletion, the Δψm showed not a". Please change it to "In contrast to AK release and ATP depletion, the Δψm did not show"
- Line 273: While MC was not toxic, 4-CMC (1000 μM) and 4-MMC (1000 μM) and MDMA (500 μM) significantly (move here) decreased mitochondrial basal and FCCP-stimulated respiration at 37 °C.
Author Response
- This study is well planned however, authors restricted their study to two different temperatures: 37°C and 40.5°C. I would recommend to use additional temp measurements of 38°C and 39°C as the amount of temperature increase is dose dependent.
Answer: We agree with the Reviewer that it would be ideal to have results from temperatures between 37°C and 40.5°C. We evaluated the possibility to perform such experiments within the 10 days that we have for answering the questions of the Reviewers. Unfortunately, we must admit that this is not feasible. Cells first have to grow before they can be treated and this takes in total more time than 10 days. Furthermore, due to the current COVID-19 situation, we have only limited access to the laboratory. Therefore, it was impossible for us to add another temperature. We hope that these arguments are acceptable.
We based the choice of the higher temperature on the clinical observation that patients with a body temperature >40°C after the ingestion of ecstasy usually have severe complications such as intravascular coagulation, rhabdomyolysis, and renal or other organ failure (Temperature 2014;1:192-200). Body temperature >41°C can result in fatalities in such patients. Body temperatures <40°C are usually quite well tolerated by patients. Based in these clinical findings, we did not expect to see a large impact on toxicity at temperatures <40°C.
In their study looking at neurotoxicity of ecstasy and ecstasy metabolites, Barbosa et al. used a similar study design (Arch Toxicol 2014;88:515-531). In this study, incubations were performed for a period of 24 or 48 h under normothermic (37°C) or hyperthermic (40°C) conditions. They found that ecstasy decreased the cellular glutathione content, but was not cytotoxic for differentiated SH-SY5Y cells up to 200 μM under normo- and hyperthermic conditions. The toxicity was driven mainly by ecstasy metabolites.
We justify the temperature that we have chosen by citing the clinical data and the study of Barbosa et al. on page 13 of the manuscript. Further, we mention the study of Barbosa et al. on page 2 and again on page 13 of the manuscript.
- The amount of dose used for in vitro study was very high. Does the author have a reasonable explanation for using such as high dose?
Answer: In comparison to their pharmacological activity, which is observed in the high nanomolar to low micromolar range, depending on the compound and the pharmacological effect considered, cytotoxicity was detected at clearly higher concentrations in the current study. Plasma concentrations reached for cathinones at non-toxic, pharmacological doses are in the low micromolar range, whereas we started to see an impairment of oxygen uptake at 200 micromolar for MC, 4-CMC and 4-MMC. A possible explanation for this discrepancy between pharmacological activity and toxicity may be that cell lines as used in the current study may be less sensitive to toxicants than primary cells. This has for instance been shown when comparing the effect of hepatotoxicants on human hepatocyte cell lines and primary human hepatocytes. Furthermore, patients presenting with neurotoxicity usually have ingested higher than pharmacological doses of these compounds and may have ingested other toxic drugs and/or alcohol. In addition, these compounds generally reach higher concentrations in the brain than in plasma (Psychopharmacology 2019;236:881-890).
Taken together, these factors may be sufficient to explain the gap between the concentrations associated with a pharmacological activity and toxic effects observed for these compounds in the current in vitro investigations.
We discuss that on page 14 of the manuscript.
- Line 248: "In contrast to AK release and ATP depletion, the Δψm showed not a". Please change it to "In contrast to AK release and ATP depletion, the Δψm did not show"
Answer: we have corrected that.
- Line 273: While MC was not toxic, 4-CMC (1000 μM) and 4-MMC (1000 μM) and MDMA (500 μM) significantly (move here) decreased mitochondrial basal and FCCP-stimulated respiration at 37°C.
Answer: we have corrected that.
Reviewer 2 Report
Well done on this nice piece of work.
It clearly offers an excellent contribution on the mechanisms underpinning morbidity and mortality in patients using NPS.
Only minor spellchecks required.
Please optimise and reduce to a minimum the number of pictures and graphs . They tend to become slightly confusing.
Author Response
- Only minor spellchecks required.
Answer: we checked the manuscript for spelling errors and corrected what we found.
- Please optimise and reduce to a minimum the number of pictures and graphs . They tend to become slightly confusing.
Answer: Please, consider also our answers to questions 5 and 8 to Reviewer 3. We have modified Fig. 4 and Fig. 6, because they were too complicated. Regarding Fig 4, we removed the non-affected cells and regarding Fig. 6, we give better explanations in the text. We hope that this is ok like that for you.
Reviewer 3 Report
Cells (Manuscript ID: cells-741931), Comments to the Authors:
Title: Hyperthermia increases neurotoxicity associated with novel methcathinones
Comments
The submitted manuscript discussed the effects of hyperthermia (40.5 °C) on the neurotoxicity of methcathinone (MC), 4-chloromethcathinone (4 CMC), and 4 methylmethcathinone (4-MMC) in SH-SY5Y cells. The authors found that 4-CMC and 4 MMC were cytotoxic (decrease in cellular ATP and plasma membrane damage) under both hyper- (40.5 °C) and normothermic conditions (37 °C), whereby cells were more sensitive to the toxicants at 40.5 °C. 4 CMC and 4-MMC impaired the function of the mitochondrial electron transport chain and increased mitochondrial formation of reactive oxygen species (ROS) in SH-SY5Y cells, which was accentuated under hyperthermic conditions. Hyperthermia was associated with a rapid expression of the 70 kilodalton heat shock protein (Hsp70), which partially prevented cell death after 6 h of exposure to the toxicants. After 24 h of exposure, autophagy was stimulated by the toxicants and by hyperthermia, but could only partially prevent cell death. In conclusion, hyperthermic conditions increased the neurotoxic properties of methcathinones despite the stimulation of protective mechanisms.
I think the manuscript can be accepted for publication after the authors respond to the following comments:
- Why do the authors have two identical figures in their submitted manuscript, Figure 1 and Figure S2?
- Why did the authors select such doses? Do they have any estimation of the dose inside the human body or just they came up with such doses in ascending order?
- Why did the authors test the effect of MC on the mitochondria membrane potential and did not test the effect of 4-FMC?
- This sentence is not clear and needs rephrasing "None of the investigated drugs stimulated the leak respiration, suggesting that there was no uncoupling of oxidative phosphorylation."
- Figure 4A is complicated and should be divided into two figures.
- The authors should comment on the observation that HSp70 expression was decreased with the use of 2000 micro mole of 4-MMC compared with 1000 micro mole.
- Why did not the authors use a positive control in HSp70 expression experiment?
- The paragraph discussing the use of concanamycin A and bafilomycin A is not clear and needs rephrasing.
- Why did not the authors use the positive control fluorescence microscopy visualization of AVOs stained with AO.
Author Response
- Why do the authors have two identical figures in their submitted manuscript, Figure 1 and Figure S2?
Answer: The two figures are not identical.
In Figure 1, we showed the AK release and ATP content of the following compounds: MC, 4-FMC, 4-CMC, and MDMA (cathinones and ecstasy).
In Figure S2, we showed the AK release and ATP content of the following compounds: Amphetamine, 4-FA, PCA (amphetamines).
Based on the findings presented in these figures, we decided to continue with the cathinones. Nevertheless, we find it important to give also the data for the amphetamines.
- Why did the authors select such doses? Do they have any estimation of the dose inside the human body or just they came up with such doses in ascending order?
Answer: We have received the same question from Reviewer 1. Our answer to this question is as follows:
In comparison to their pharmacological activity, which is observed in the high nanomolar to low micromolar range, depending on the compound and the pharmacological effect considered, cytotoxicity was detected at clearly higher concentrations in the current study. Plasma concentrations reached for cathinones at non-toxic, pharmacological doses are in the low micromolar range, whereas we started to see an impairment of oxygen uptake at 200 micromolar for MC, 4-CMC and 4-MMC. A possible explanation for this discrepancy between pharmacological activity and toxicity may be that cell lines as used in the current study may be less sensitive to toxicants than primary cells. This has for instance been shown when comparing the effect of hepatotoxicants on human hepatocyte cell lines and primary human hepatocytes. Furthermore, patients presenting with neurotoxicity usually have ingested higher than pharmacological doses of these compounds and may have ingested other toxic drugs and/or alcohol. In addition, these compounds generally reach higher concentrations in the brain than in plasma (Psychopharmacology 2019;236:881-890).
Taken together, these factors may be sufficient to explain the gap between the concentrations associated with a pharmacological activity and toxic effects observed for these compounds in the current in vitro investigations.
We discuss that on page 14 of the manuscript.
- Why did the authors test the effect of MC on the mitochondria membrane potential and did not test the effect of 4-FMC?
Answer: As stated in manuscript, after the initial experiments, we first decided to go on with the cathinones and then, we decided to study: MC, 4-CMC, 4-MMC. The restriction to 3 cathinones was also due to capacity reasons.
4-CMC and 4-MMC were selected as the compounds with the largest difference between the two temperature conditions in the initial experiments (AK release and cellular ATP content). In comparison, 4-FMC did not reach a significant difference in these experiments. MC was selected despite its low toxicity because it can be regarded as the parent compound of 4-CMC and 4-MMC.
- This sentence is not clear and needs rephrasing "None of the investigated drugs stimulated the leak respiration, suggesting that there was no uncoupling of oxidative phosphorylation."
Answer: The leak respiration is the respiration in the presence of oligomycin, which blocks the F1F0-ATP synthase or mitochondrial ATP production. Oxygen uptake in the presence of oligomycin therefore represents use of oxygen without mitochondrial ATP production, this means uncoupling of oxygen consumption and production of ATP. Since we did not observe an increase in the leak respiration, we excluded uncoupling of oxidative phosphorylation as a reason for the reduced mitochondrial membrane potential.
We give more explanation and rephrased the sentence on page 7.
- Figure 4A is complicated and should be divided into two figures.
Answer: we agree with the Reviewer that there is much information in Fig. 4A. Nevertheless, we find it better to give the important results in one figure and not to split. Since we focus on cell death in this figure, we decided to remove the part of the columns with the viable cells. We believe that the figure is less busy like that and should be better understandable.
- The authors should comment on the observation that Hsp70 expression was decreased with the use of 2000 micro mole of 4-MMC compared with 1000 micro mole.
Answer: There was a numerical decrease in Hsp70 expression in the presence of 2000 µM 4-CMC and 4-MMC, whereas 2000 µM MC was associated with a numerical increase compared to 1000 µM. As mentioned, the changes were only numerical, without statistical significance and there was no significant difference to control incubations. This means that observed increases in Hsp70 expression can be explained by the rise in temperature, as discussed in the manuscript.
- Why did not the authors use a positive control in HSp70 expression experiment?
Answer: For us, the condition “DMSO 0.1% at 40.5°C” was the positive control for the induction of the expression of Hsp70 (versus the situation at 37°C).
- The paragraph discussing the use of concanamycin A and bafilomycin A is not clear and needs rephrasing.
Answer: We agree with the Reviewer that this paragraph was difficult to understand. We have rephrased the entire paragraph on page 12 of the revised manuscript. It is important to realize that concanamycin A and bafilomycin A impair autophagosome removal and that in Fig. 6B the LC3 II/LC3 I ratios of incubations containing only concanamycin A or bafilomycin A and of incubations containing concanamycin A or bafilomycin A in combination with 4 CMC or 4 MMC were not different at both temperatures. Since, as shown in Fig. 6A, the LC3 II/LC3 I ratio was increased by 4-CMC and 4-MMC in the absence of concanamycin A and bafilomycin A, we conclude that 4-CMC and 4-MMC mainly impair the degradation of the autophagosomes. We made that clear with the rephrasing of the paragraph.
- Why did not the authors use the positive control fluorescence microscopy visualization of AVOs stained with AO.
Based on the results of Figure 6, where we showed the first steps in the formation of autophagosome (LC3 II/LC3 I ratio), we then proceeded to the visualization and quantification of the late stages in the autophagic process. In order to quantify acidic vesicular organelles (AVOs), we exposed cells to acridine orange (AO) and determined the red fluorescence as described in the Method section using flow cytometry. In our view and experience, flow cytometry is a reliable method for quantification and yields reproducible results as shown in Fig. 7B.
Round 2
Reviewer 3 Report
Cells (Manuscript ID: cells-741931), Comments to the Authors:
Title: Hyperthermia increases neurotoxicity associated with novel methcathinones
Comments
After reading the authors' response to my comments, I think the authors responded to all my remarks and I think the manuscript can be accepted for publication.